# Quantum-Inspired Image Encodings for Financial Time-Series Forecasting

## Abstract

This study proposes a quantum-inspired methodology that transforms time-series data into image representations for prediction. Unlike classical encodings such as the Gramian Angular Field (GAF), Recurrence Plot (RP), and Markov Transition Field (MTF), which rely on additive pairwise relations, our approach embeds both probabilistic amplitudes and dynamic phase information. Observations are first mapped into quantum amplitudes via Gaussian soft encoding, and local temporal structures are incorporated through phase-function encoding, allowing interference effects that reveal volatility, cumulative imbalances, and phase shifts hidden to classical methods. Building on this foundation, we extend GAF, RP, and MTF into their quantum analogues—Q-GAF, Q-RP, and Q-MTF—producing complex-domain images suitable for CNN-based forecasting. Empirical analysis on the S&P 500 and Russell 3000 indices shows that these quantum-inspired encodings substantially improve predictive accuracy. Our contributions are both methodological and empirical: we present a novel representation framework for financial time series and demonstrate that quantum-inspired image encodings capture richer dynamics and previously undetectable patterns, with implications for forecasting and risk modeling.

## 1 Introduction And Related Works

Time-series classification (TSC) has become increasingly important across domains where accurate recognition of temporal patterns carries both scientific and practical significance. In finance in particular, improved predictive models translate directly into better decision-making and economic value.

The rapid progress of deep learning reshapes how such problems are addressed. Convolutional neural networks (CNNs), which achieved groundbreaking success in image recognition tasks (Krizhevsky et al., 2012; LeCun et al., 2010), have been adapted to time-series settings by transforming one-dimensional signals into two-dimensional representations. Literature shows that treating time series as texture-like images enables CNNs to extract discriminative features that remain inaccessible in raw sequential form.

Well-established approaches convert time series into images, chief among them are Gramian Angular Fields (GAF) and Markov Transition Fields (MTF) introduced by (Wang & Oates, 2015), and Recurrence Plots (RP) by Hatami et al. (2018). These approaches introduce new feature types beyond 1D signals, enable intuitive visualization of periodic and trajectory characteristics, and have achieved competitive accuracy. Importantly, in financial forecasting, encoding market time series into GAF images has been shown to improve the prediction of market behavior Barra et al. (2020). These developments confirm that image encoding constitutes a powerful approach for extracting and leveraging complicated structures in time-series data. Despite these advances, existing image encodings remain fundamentally constrained by their reliance on real-valued, additive formulations. This limitation motivates a richer representation space—one capable of capturing both probabilistic amplitudes and phase dynamics.

Time-series data are not merely sequences of real values; they often embody probabilistic and phase-like structures that classical encodings systematically overlook. In this respect, a quantum-inspired representation provides a more suitable framework for capturing and expressing such dynamics (e.g., (Ahn et al., 2018)). From the perspective of representation, existing image-encoding methods are

grounded in real-valued formulations that reduce pairwise relationships to additive combinations without phase information. By contrast, when a time series is represented as a quantum-inspired state, the combination of complex amplitudes and phases enables constructive or destructive interference, thereby revealing patterns that classical additive statistics cannot detect. In this sense, quantum encoding goes beyond a metaphor drawn from physics: it opens a fundamentally richer functional space through complex-valued, phase-aware representations.

Financial time series, such as stock prices, exemplify why such an enriched framework is needed. Their defining characteristics include volatility clustering, where variance structures evolve even when the mean remains constant, asymmetric accumulation effects between upward and downward movements, and the interplay between short-term noise and long-term mean reversion Bondt & Thaler (1989). The phase functions we introduce—whether expectation-to-volatility or volatility-based—map naturally onto these empirical properties, allowing the encoding to reflect precisely those structural asymmetries and dynamics. Thus, our approach does not simply borrow the language of quantum mechanics; it leverages the analogy as a mathematically coherent means of representing the probabilistic and phase-like structures intrinsic to financial data.

Finally, the adoption of a quantum formalism carries practical significance for predictive performance. Classical encodings, such as GAF, RP, and MTF, struggle to capture hidden nonlinear patterns, including volatility transitions and phase shifts, which limit their forecasting power. By contrast, quantum-inspired encodings are capable of uncovering such latent dynamics, transforming the framework from a conceptual embellishment into a source of measurable gains in predictive accuracy. This dual role—conceptual richness and empirical effectiveness—underlines the substantive necessity of adopting quantum analogies in time-series forecasting.

Methodologically, we extend classical GAF, RP, and MTF into their quantum analogues—Q-GAF, Q-RP, and Q-MTF—generating images from quantum states. Our framework maps normalized time-series values into probability amplitudes over discretized basis intervals through Gaussian soft encoding, embedding the distributional structure of the data. Local dynamics are then incorporated via phase-function encoding based on local windows, which reflects features such as cumulative imbalance and volatility. The resulting complex-valued states combine amplitudes and phases, allowing interference effects that classical encodings cannot capture. Building on these states, we construct quantum-inspired extensions of GAF, RP, and MTF, yielding image representations that integrate both probabilistic and dynamical information for forecasting with CNNs.

To evaluate the empirical effectiveness of our framework, we conduct experiments on two representative equity indices—the Russell 3000 and the S&P 500—over the period from 2009 through the first quarter of 2025. A moving-window procedure is employed, with two years of training data, six months for validation, and three months for testing in each iteration. Across this extensive evaluation, all quantum-analogue models consistently outperform their classical counterparts, achieving on average 2.6% higher accuracy. Under the same evaluation procedure, the average win rate across the quantum models exceed that of the classical benchmarks by 32.9%. Furthermore, the best-performing quantum variant further surpass the strongest classical baseline, yielding gains of 41.1% in average win rate. Taken together, these findings provide strong empirical evidence that quantum-inspired image encodings offer a more suitable and effective representation for forecasting financial time series.

The contributions of this study can be summarized along two dimensions. First, on the methodological side, we introduce a novel quantum-inspired framework that encodes time-series data into complex states combining probabilistic amplitudes and phase information, and extend classical image representations into their quantum analogues. Second, on the empirical side, we demonstrate that these quantum-inspired representations consistently outperform their classical counterparts in forecasting accuracy. Taken together, these findings provide strong empirical evidence that quantum-inspired image encodings offer a more effective representation for financial time series forecasting, including latent structures and dynamics that remain undetected by classical approaches.

## 2 QUANTUM STATE ENCODING

Assume the original time-series is as follows:

$$\mathbb{X}_{\text{origin}} = \{x_1, x_2, \ldots, x_T\} \quad \text{where } x_t \in \mathbb{R}.$$

After normalization,

$$\mathbb{X}_{\text{norm}} = \left\{ x \;\middle|\; x = \frac{x_{\text{origin}} - \mu_{\mathbb{X}_{\text{origin}}}}{\sigma_{\mathbb{X}_{\text{origin}}}}, \; x_{\text{origin}} \in \mathbb{X}_{\text{origin}} \right\}$$

$$\mu_{\text{global}} = \frac{1}{T} \sum_{t=1}^{T} x_t, \quad \sigma_{\text{global}} = \sqrt{\frac{1}{T} \sum_{t=1}^{T} (x_t - \mu)^2} \quad \text{where } x_t \in \mathbb{X}_{\text{norm}}.$$

## 2.1 GAUSSIAN SOFT ENCODING

To represent normalized time-series values as quantum states, we first apply a soft encoding procedure. This encoding incorporates both the magnitude and probabilistic information of the quantum state, capturing the probabilistic weights that determine the likelihood of a given time-series observation falling into each interval. The subsequent step formalizes these intervals through a precise bin definition.

### 2.1.1 DEFINITION OF QUANTUM STATE BIN SET

$$\text{Bin range} = \left[\, \mu_{\text{global}} - k\sigma_{\text{global}}, \; \mu_{\text{global}} + k\sigma_{\text{global}} \,\right]$$

$$\text{The bin interval } \Delta b = \frac{\sigma_{\text{global}}}{h}$$

where $k$ is a hyperparameter that controls the bin range in multiples of $\sigma_{\text{global}}$, and $h$ is a constant that determines the bin interval so that the number of bins $N_{\text{bins}} = 2kh + 1$. In this work, we set $k = 3$ and $h = 10$ as a baseline configuration, which provides a reasonable trade-off between resolution and computational cost.

The bin set is defined as follows:

$$\mathbb{B} = \{\, b_m \mid b_m = \mu_{\text{global}} - k\sigma_{\text{global}} + (m-1)\,\Delta b, \; m = 1, \ldots, N_{\text{bins}} \,\}.$$

$$\text{Each bin center defines a quantum state } |b_m\rangle$$

### 2.1.2 UNNORMALIZED GAUSSIAN WEIGHT

The unnormalized Gaussian weight function serves as a distance-based Gaussian measure, reflecting how strongly $x_t$ is associated with the bin center $b_m$. The function is as follows:

$$\tilde{P}_m(x_t) = \exp\left[-\frac{(x_t - b_m)^2}{2\sigma_{\text{kernel}}^2}\right], \quad \text{where } \sigma_{\text{kernel}} = \sigma_{\text{scale}} \cdot \sigma_{\text{global}}.$$

The scaling factor $\sigma_{\text{scale}}$ is a hyperparameter to adjust the kernel width; we set $\sigma_{\text{scale}} = 1$ as the baseline in this work.

### 2.1.3 NORMALIZED PROBABILITY DISTRIBUTION OVER BINS

$$P_m(x_t) = \frac{\tilde{P}_m(x_t)}{\sum_{j=1}^{N_{\text{bins}}} \tilde{P}_j(x_t)} \quad \text{so that} \sum_{m=1}^{N_{\text{bins}}} P_m(x_t) = 1$$

$$\mathbb{P}(x_t) = \langle P_1(x_t), P_2(x_t), \ldots, P_{N_{\text{bins}}}(x_t) \rangle$$

Here, $P(x_t)$ denotes the probability vector representing the likelihood that a specific $x_t$ belongs to each bin $b_m$.

### 2.1.4 PROBABILITY AMPLITUDE MAPPING

As the quantum amplitude corresponds to the square root of the probability, we can define the amplitude of each bin $b_m$ for $x_t$ as follows:

$$\alpha_m(x_t) = \sqrt{P_m(x_t)}.$$

Then, without the phase information applied yet, the quantum state at time $t$ can be represented as a superposition of the basis states $|b_m\rangle$:

$$|\psi_t\rangle = \sum_m \alpha_m(x_t) |b_m\rangle$$

## 2.2 PHASE FUNCTION ENCODING

There are four proto-types that we use as a phase function $\phi_m(t)$, which captures the intrinsic dynamical properties of the time series. We first define the local time window centered at $t$ as

$$W_t = \{x_{t-w}, \ldots, x_{t-1}, x_t, x_{t+1}, \ldots, x_{t+w}\}$$

where $x_t \in \mathbb{X}_{\text{norm}}$ and the window size is $2w + 1$. To ensure empirical practicality, we set the default window size to the larger of 3 and the smallest integer not less than $10\%$ of the total length of the time series, $|X_{\text{origin}}|$. Moreover, to maintain symmetry around the center point, if the resulting window size is even, we increment it by 1. We further define the sets of positive (non-negative) and negative signal values within the window as

$$U_t = \{x \in W_t \mid x \geq 0\}, \quad D_t = \{x \in W_t \mid x < 0\}.$$

### 2.2.1 EXPECTATION-TO-VOLATILITY RATIO PHASE

$$\phi_m^{\text{exvol}}(t) = \begin{cases} \pi \cdot \text{sigmoid}\left(\lambda_e \cdot \dfrac{E_{\text{up}}(t)}{\sigma_{\text{global}}}\right), & b_m > 0, \\ \pi \cdot \text{sigmoid}\left(\lambda_e \cdot \dfrac{E_{\text{down}}(t)}{\sigma_{\text{global}}}\right), & b_m < 0, \\ 0, & \text{otherwise}, \end{cases}$$

where

$$\text{sigmoid}(z) = \frac{1}{1+e^{-z}}, \quad E_{\text{up}}(t) = \frac{1}{|U_t|}\sum_{x \in U_t} x, \quad E_{\text{down}}(t) = \frac{1}{|D_t|}\sum_{x \in D_t} x,$$

and $\lambda_e$ is a scaling hyperparameter, set to 1 as our baseline.

### 2.2.2 VOLATILITY PHASE

$$\phi_m^{\text{vol}}(t) = \pi \cdot \frac{\sigma_{t,\text{local}}}{\lambda_v \cdot \sigma_{\text{global}}},$$

where $\lambda_v$ is a scaling hyperparameter, set to 3 as our baseline.

## 2.3 FINAL QUANTUM STATE

Soft encoding conveys the magnitude of the state through probabilistic weights, while phase encoding enriches it with temporal and structural characteristics, so that the final state integrates both amplitude and phase. Including both the probabilistic amplitude and the dynamic phase information, the quantum state at time $t$ can be expressed as a superposition of the basis states $|b_m\rangle$. In our framework, a quantum state at time $t$ can be expressed in two equivalent forms. First, using the standard ket notation:

$$|\psi_t\rangle = \sum_{m=1}^{N_{\text{bins}}} \alpha_m(x_t)\, e^{i\phi_m(t)} |b_m\rangle,$$

where $\{|b_m\rangle\}_{m=1}^{N_{\text{bins}}}$ denotes the orthonormal basis states.

Equivalently, this state can be represented as a complex vector in $\mathbb{C}^{N_{\text{bins}}}$:

$$\psi_t \equiv (\psi_{t,1}, \psi_{t,2}, \ldots, \psi_{t,N_{\text{bins}}}) \equiv \begin{bmatrix} \alpha_1(x_t)e^{i\phi_1(t)} \\ \alpha_2(x_t)e^{i\phi_2(t)} \\ \vdots \\ \alpha_{N_{\text{bins}}}(x_t)e^{i\phi_{N_{\text{bins}}}(t)} \end{bmatrix} \in \mathbb{C}^{N_{\text{bins}}}.$$

Thus, the ket notation $|\psi_t\rangle$ and the coordinate representation $\psi_t \in \mathbb{C}^{N_{\text{bins}}}$ are mathematically equivalent descriptions of the same quantum state.

## 3 COMPLEX DOMAIN IMAGE ENCODING

The time-series information expressed as quantum states is further encoded into images to enable application to AI-based image classification models such as CNNs. We extend the classical real-valued approaches of Gramian Angular Field (GAF), Recurrence Plot (RP), and Markov Transition Field (MTF) into the complex domain, thereby proposing advanced methodologies that capture richer temporal dynamics.

### 3.1 QUANTUM-GAF

Quantum-GAF (Q-GAF) encodes temporal correlations by combining both the amplitude and phase of quantum states. Unlike the classical GAF, which only relies on rescaled real values and angles, Q-GAF incorporates complex amplitudes, capturing richer dynamics of the time series.

#### 3.1.1 ALPHA-WEIGHTED

$$G_{t_i,t_j} = \sum_{m=1}^{N_{\text{bins}}} \alpha_m(x_{t_i}) \, \alpha_m(x_{t_j}) \, \cos\big(\phi_m(t_i) + \phi_m(t_j)\big)$$

where
$t_i, t_j \in \{1, \ldots, T\}$: time indices,
$m \in \{1, \ldots, N_{\text{bins}}\}$: bin index,
$\alpha_m(x_t) = |\psi_{t,m}|$: probability amplitude at bin $m$ for $x_t$,
$\phi_m(t) = \arg(\psi_{t,m})$: phase at bin $m$.

#### 3.1.2 REAL INNER-PRODUCT BASED

$$G_{t_i,t_j} = \Re\big(\psi_{t_i} \, \psi_{t_j}^*\big) \quad \text{(scalar state)}$$

$$G = \Re\big(\Psi\Psi^\dagger\big) \quad \text{(vector state)}$$

where
$\psi_t \in \mathbb{C}^{N_{\text{bins}}}$: quantum state vector at time $t$,
$\psi_{t_j}^*$: complex conjugate of $\psi_{t_j}$,
$\Psi \in \mathbb{C}^{T \times N_{\text{bins}}}$: stacked quantum states matrix for all time,
$\Psi^\dagger$: Hermitian transpose (conjugate transpose) of $\Psi$,
$\Re(\cdot)$: real part.

### 3.2 QUANTUM-RP

Quantum-RP (Q-RP) measures recurrence structures based on quantum state similarity, either by distance or fidelity. Compared to the classical RP, which uses raw value differences, Q-RP leverages complex-valued amplitudes and phases, providing a more informative recurrence map.

### 3.2.1 L2-NORM BASED

$$RP_{t_i,t_j} = \|\psi_{t_i} - \psi_{t_j}\|_2 = \sqrt{\sum_{m=1}^{N_{\text{bins}}} \left|\psi_{t_i,m} - \psi_{t_j,m}\right|^2}$$

where
$\psi_t \in \mathbb{C}^{N_{\text{bins}}}$: quantum state vector at time $t$,
$\psi_{t,m}$: $m$-th bin component of $\psi_t$,
$|\cdot|$: complex modulus,
$\|\cdot\|_2$: Euclidean norm (always real and nonnegative).

### 3.2.2 FIDELITY BASED

$$RP_{t_i,t_j} = \left|\langle \psi_{t_i} \mid \psi_{t_j} \rangle\right|^2 = \left|\sum_{m=1}^{N_{\text{bins}}} \psi_{t_i,m}^* \psi_{t_j,m}\right|^2$$

where
$\langle \psi_{t_i} \mid \psi_{t_j} \rangle = \sum_{m=1}^{N_{\text{bins}}} \psi_{t_i,m}^* \psi_{t_j,m}$: inner product,
$\psi_{t_i,m}^*$: complex conjugate of $\psi_{t_i,m}$,
$|\cdot|^2$: squared magnitude (real, $\geq 0$) where 1 if identical, 0 if orthogonal.

### 3.3 QUANTUM-MTF

Quantum-MTF (Q-MTF) discretizes quantum states via clustering and encodes their transition probabilities across time. While classical MTF is built on discretized raw values, Q-MTF captures transitions between complex quantum states, reflecting both amplitude and phase evolution.
Complex states are mapped to real vectors by concatenating real and imaginary parts as follows:

$$\tilde{\psi}_t = \left(\Re(\psi_t), \Im(\psi_t)\right) \in \mathbb{R}^{2N_{\text{bins}}}$$

Then, using K-means clustering, each $\tilde{\psi}_t$ is assigned to a discrete state (cluster label):

$$q_t \in \{1, 2, \ldots, K\}, \quad t = 1, \ldots, T$$

where $K$ is the hyperparameter to decide the number of clusters and we tentatively choose 5 as a baseline. Then, to count transitions between two discrete states, we define the transition counts from a to b:

$$C_{ab} = \#\{\, t \mid q_t = a,\ q_{t+1} = b \,\}, \quad a, b \in \{1, 2, \ldots, K\}$$

and the normalized transition probability matrix from a to b:

$$W_{ab} = \frac{C_{ab}}{\sum_{b'=1}^{K} C_{ab'}}, \quad W \in [0, 1]^{K \times K}.$$

Finally, the Q-MTF image matrix $M$ is obtained by indexing $W$ with the state labels of each time step as follows:

$$M_{t_i,t_j} = W_{q_{t_i}, q_{t_j}}, \quad t_i, t_j \in \{1, \ldots, T\}.$$

## 4 EXPERIMENTS

### 4.1 SETTINGS

For the empirical study, we use high-frequency equity data from two representative U.S. market indices—the Russell 3000 and the S&P 500—at five-minute intervals. The sample covers the period from January 2, 2009, to April 1, 2025. We retain only those trading days with at least 30 observations recorded at five-minute intervals, and in order to ensure a uniform time length across days, we restrict the intraday window to 08:00–17:00. Missing values are imputed by linear interpolation, defined as the average of the two nearest neighboring observations.

Table 1: Comparative performance of quantum-inspired and classical encodings

| Metric | Dataset | Q-GAF (4) | | | | Q-MTF (2) | | | | Q-RP (4) | | Classical (4) | | | |
|---|---|---|---|---|---|---|---|---|---|---|---|---|---|---|---|
| | | Alpha-exvol | Alpha-vol | Inner-exvol | Inner-vol | exvol | vol | L2-exvol | L2-vol | Fidelity-exvol | Fidelity-vol | GAF (GADF) | GAF (GASF) | MTF | RP |
| Accuracy | Russell3000 | 0.512 | 0.540 | 0.514 | 0.539 | 0.499 | 0.499 | 0.507 | 0.517 | 0.507 | 0.505 | 0.492 | 0.503 | 0.483 | 0.505 |
| | | (0.070) | (0.063) | (0.075) | (0.073) | (0.057) | (0.058) | (0.068) | (0.064) | (0.079) | (0.083) | (0.065) | (0.050) | (0.066) | (0.069) |
| | S&P500 | 0.524 | 0.544 | 0.516 | 0.543 | 0.519 | 0.503 | 0.518 | 0.514 | 0.508 | 0.514 | 0.532 | 0.506 | 0.502 | 0.507 |
| | | (0.076) | (0.059) | (0.082) | (0.071) | (0.051) | (0.078) | (0.070) | (0.086) | (0.071) | (0.067) | (0.067) | (0.070) | (0.066) | (0.074) |
| **Among whole models** | | | | | | | | | | | | | | | |
| Total Wins | | 8 | 7 | 6 | 8 | 4 | 6 | 3 | 7 | 1 | 9 | 5 | 3 | 3 | 2 |
| Avg. Rank | | 6.98 | 4.96 | 6.91 | 5.38 | 7.33 | 8.00 | 7.11 | 6.53 | 6.93 | 7.27 | 6.91 | 7.69 | 7.95 | 7.67 |
| Avg. Wins (%) | | 14.5 | 12.7 | 10.9 | 14.5 | 7.3 | 10.9 | 5.5 | 12.7 | 1.8 | 16.4 | 9.1 | 5.5 | 5.5 | 3.6 |
| **Among the same baseline models** | | | | | | | | | | | | | | | |
| Total Wins | | 11 | 14 | 9 | 17 | 24 | 21 | 12 | 16 | 10 | 17 | 6 | 17 | 12 | |
| Avg. Rank | | 3.62 | 2.60 | 3.40 | 2.80 | 1.80 | 1.98 | 2.84 | 2.66 | 2.82 | 2.89 | 3.51 | 3.89 | 2.02 | 3.00 |
| Avg. Wins (%) | | 20.0 | 25.5 | 16.4 | 30.9 | 43.6 | 38.2 | 21.8 | 29.1 | 18.2 | 30.9 | 18.2 | 10.9 | 30.9 | 21.8 |

The dependent variable is constructed as a binary indicator that takes the value 1 if the next day's closing price exceeds its opening price, and 0 otherwise, thereby capturing whether the market rises on the following day. To evaluate forecasting performance, we adopt a rolling-window procedure with six-month steps. In each iteration, the estimation sample consists of a two-year training period, followed by a six-month validation set, and a three-month test set.

We employ a CNN consisting of two convolutional layers with 16 and 32 filters of size 3×3, each followed by a 2×2 max-pooling layer. The extracted feature maps are flattened and passed through a fully connected layer with 64 units and ReLU activation. To reduce overfitting, a dropout rate of 0.3 is applied. Finally, the output layer consists of N units with a softmax/sigmoid activation, depending on the classification task. The model was trained for 10 epochs with a batch size of 8.

## 4.2 RESULT AND DISCUSSION

Table 1 shows the comparative performance of the proposed quantum-inspired models against their classical counterparts where the total number of shifted windows is 28. In this table, *"Among whole models"* refers to evaluations conducted across the entire set of models, while *"Among the same baseline models"* restricts the comparison to each quantum variant and its corresponding classical baseline. The notations *"exvol"* and *"vol"* indicate the phase functions based on the *expectation-to-volatility ratio* and *volatility*, respectively. Reported values in parentheses under *Accuracy* denote standard deviations. For the classical models, the entries *GASF* and *GADF* correspond to the Gramian Angular Summation Field (GASF), which uses the sum of angular values to represent temporal correlations, and the Gramian Angular Difference Field (GADF), which relies on angular differences, respectively.

The empirical results reveal a consistent performance advantage of the quantum-inspired encodings. Across the full evaluation, the quantum models deliver an average accuracy improvement of 2.6% over classical benchmarks. When restricting the comparison to identical baselines, their average win rate exceeds that of the classical models by 32.9%. Moreover, the strongest quantum variant outperforms the best classical method, achieving additional gains of 41.1% in win rate. These findings provide robust evidence that quantum-inspired image encodings not only capture richer temporal dynamics but also yield practical forecasting gains beyond the reach of conventional methods.

## 5 CONCLUSIONS

This paper presents a quantum-inspired framework for transforming financial time-series data into complex-domain image representations. Methodologically, we develop Gaussian soft encoding and phase-function encoding to embed both probabilistic amplitudes and temporal dynamics, and extend classical GAF, RP, and MTF into their quantum analogues. These innovations expand the representational space available for time-series modeling beyond the additive and real-valued constraints of conventional approaches.

Empirically, extensive experiments using high-frequency data from the S&P 500 and Russell 3000 indices demonstrate that the proposed quantum-inspired encodings deliver consistent forecasting improvements. Relative to established baselines, they achieve on the order of 2.6% higher accuracy and a 32.9% advantage in average win rate, with the strongest quantum variant exceeding the best classical benchmark by 41.1% in average win rate. These results highlight the practical value of

encoding amplitude–phase structures, capturing latent temporal dynamics and market phase patterns that classical models fail to detect.

Despite these findings, some limitations remain. Our models require greater computational resources than conventional encodings, and the present evaluation is restricted to U.S. equity indices. Future work may explore the scalability of quantum-inspired encodings to larger universes of assets, other macroeconomic domains, and integration with advanced architectures beyond CNNs, such as complex-valued deep networks. Addressing these directions may further clarify the scope and robustness of quantum-inspired image representations in time-series forecasting.

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
