# OpenReview forum: "Quantum-Inspired Image Encodings for Financial Time-Series Forecasting"
_ICLR.cc/2026/Conference — ICLR 2026 Conference Withdrawn Submission_

### Official Review · Reviewer_6Coq · 2025-10-31

**Soundness:** 1
**Presentation:** 1
**Contribution:** 2
**Rating:** 2
**Confidence:** 4

**Summary:**

This paper introduces a novel "quantum-inspired" framework for financial time-series forecasting. The core idea is to encode one-dimensional time-series data into two-dimensional, complex-valued image representations, which are then fed into a Convolutional Neural Network (CNN) for prediction. Specifically, the authors extend classical image-encoding methods like Gramian Angular Fields (GAF), Recurrence Plots (RP), and Markov Transition Fields (MTF) into their "quantum analogues" (Q-GAF, Q-RP, and Q-MTF) by incorporating probabilistic amplitudes and dynamic phase information borrowed from the mathematics of quantum mechanics. The primary contribution is methodological: creating a richer representation space that can capture complex dynamics missed by traditional methods, supported by empirical evidence of its effectiveness.

**Strengths:**

Applying the mathematical formalism of quantum mechanics to feature engineering for financial time series is a highly creative and interesting approach. It moves beyond conventional real-valued representations, offering new possibilities for capturing latent data structures, such as volatility and phase dynamics, by leveraging complex-valued spaces.

The paper presents an extensive empirical study on two major equity indices. The proposed quantum-inspired models consistently outperform their classical counterparts in terms of both predictive accuracy and average win rate, providing strong initial evidence for the practical utility of the framework.

**Weaknesses:**

This paper fails to provide the necessary background for non-specialists and fails to offer a clear, reproducible algorithm for specialists. This makes the paper's contribution difficult to evaluate and seriously compromises its quality as a rigorous scientific publication.

The paper successfully maps time-series data points to probability amplitudes and phases but lacks a deep theoretical justification for why this specific analogy is appropriate and superior.

The experiments effectively demonstrate the superiority of the proposed methods over their direct classical counterparts. However, to establish the true value and novelty of this encoding framework, it must be benchmarked against a broader set of state-of-the-art time-series forecasting models.

**Questions:**

See weakness

---

### Official Review · Reviewer_Jsfq · 2025-10-31

**Soundness:** 3
**Presentation:** 2
**Contribution:** 2
**Rating:** 4
**Confidence:** 4

**Summary:**

The paper proposes a quantum-inspired method for financial time series that transforms one-dimensional signals into two-dimensional images for forecasting. This transformation also enables classical encodings to suit CNN-based forecaster. Using five-minute U.S. equity data, such as S&P 500 and Russell 3000, over 2009–2025, the authors report higher classification accuracy and win rates relative to classical encodings.

**Strengths:**

- The amplitude–phase formalism extends classical image encodings and may capture salient features of financial time series.

- The paper addresses the limitations of classical encodings by lifting them into the complex domain.

- On the reported tasks and CNN architecture, quantum-inspired encodings deliver consistent accuracy and win-rate improvements over classical counterparts.

**Weaknesses:**

- The paper claims that the phase-function encoding reflects cumulative imbalance and volatility, but do not empirically verify this.

- Classification accuracy and “win rate” reported in the paper are not directly connected to monetizable alpha. Without this evaluation, it is unclear whether the accuracy uplift converts into practical use.

- Key hyperparameter choices lack sensitivity analyses.

**Questions:**

- Can you report critical finance metrics (e.g., IC, Rank IC, and Sharpe ratio) from portfolio backtests?

- How sensitive are results to hyperparameters, such as k?

- Do the phase functions measurably capture cumulative imbalance and volatility?

- How does the method perform in tests beyond 2009-2025?

- Can the quantum-inspired method generalize to other assets (e.g., individual equities, commodity futures, FX)?

---

### Official Review · Reviewer_zuHu · 2025-11-01

**Soundness:** 2
**Presentation:** 2
**Contribution:** 2
**Rating:** 4
**Confidence:** 3

**Summary:**

This paper proposes a quantum-inspired representation learning framework for financial time-series forecasting. It maps 1D normalized observations into probabilistic amplitudes and injects local temporal structure through phase-function encoding. The proposed approach outperforms the other baselines on two financial datasets.

**Strengths:**

- The quantum-inspired representation of time series is novel, introducing some unexplored features that seem useful in time series forecasting.

- The proposed approach achieves solid improvements compared to classical image encodings in experiments.

- The approach works with standard CNN-based models, suggesting that advanced CNN approaches could be explored to further enhance time series forecasting.

**Weaknesses:**

- This paper does not compare the image encoding-based methods to those without image encodings.

- Despite its effectiveness, results rely on a single CNN backbone. There is no investigation across diverse forecasting models.

- Ablation study is needed to confirm the effectiveness of each component in the proposed approach.

**Questions:**

- Can you compare the image encoding-based approaches to non-image baselines?

- Do the performance gains of quantum encodings hold for other forecasters, such as Transformer-based models?

- How do you interpret the features captured by the two components in the model?

- How do you determine the hyperparameter $k$? Do model performance totally degrade using other settings?

- What’s the complexity (or runtime) compared to baselines?


A small typo: Citation formatting at line 045 is incorrect.

---

### Note · Authors · 2025-11-21

I have read and agree with the venue's withdrawal policy on behalf of myself and my co-authors.